# SARS-CoV-2 Seropositivity in Nursing Home Staff and Residents during the First SARS-CoV-2 Wave in Flanders, Belgium

**DOI:** 10.3390/v16091461

**Published:** 2024-09-14

**Authors:** Liselore De Rop, Hanne Vercruysse, Ulysse Alenus, Judith Brusselmans, Steven Callens, Maud Claeys, Nimphe De Coene, Peter Persyn, Elizaveta Padalko, Stefan Heytens, Jan Y. Verbakel, Piet Cools

**Affiliations:** 1LUHTAR, Department of Public Health and Primary Care, KU Leuven, 3000 Leuven, Belgium; jan.verbakel@kuleuven.be; 2Research and Analytics, Liantis Occupational Health Services, 8000 Bruges, Belgium; hanne.vercruysse@liantis.be; 3Department of Diagnostic Sciences, Faculty of Medicine and Health Sciences, Ghent University, 9000 Ghent, Belgium; ulysse.alenus@gmail.com (U.A.); judith.brusselmans@ugent.be (J.B.);; 4Department of Human Structure and Repair, Faculty of Medicine and Health Sciences, Ghent University, 9000 Ghent, Belgium; 5Department of Internal Medicine & Infectious Diseases, Ghent University Hospital, 9000 Ghent, Belgium; steven.callens@uzgent.be; 6Department of Public Health and Primary Care, Faculty of Medicine and Health Sciences, Ghent University, 9000 Ghent, Belgium; nimphe.decoene@hotmail.com (N.D.C.); stefan.heytens@ugent.be (S.H.); 7Medical Department, Korian Belgium NH, 2550 Kontich, Belgium; peter.persyn@korian.be; 8Laboratory of Clinical Biology, University Hospital Ghent, 9000 Ghent, Belgium; elizaveta.padalko@ugent.be; 9Department of Diagnostic Sciences, Ghent University, 9000 Ghent, Belgium

**Keywords:** nursing homes, COVID-19, SARS-CoV-2, seroprevalence, RT-PCR testing, reinfection

## Abstract

(1) Background: early in the COVID-19 pandemic, reverse transcription polymerase chain reaction (RT-PCR) testing was limited. Assessing seroprevalence helps understand prevalence and reinfection risk. However, such data are lacking for the first epidemic wave in Belgian nursing homes. Therefore, we assessed SARS-CoV-2 seroprevalence and cumulative RT-PCR positivity in Belgian nursing homes and evaluated reinfection risk. (2) Methods: we performed a cross-sectional study in nine nursing homes in April and May 2020. Odds ratios (ORs) were calculated to compare the odds of (re)infection between seropositive and seronegative participants. (3) Results: seroprevalence was 21% (95% CI: 18–23): 22% (95% CI: 18–25) in residents and 20% (95% CI: 17–24) in staff. By 20 May 2020, cumulative RT-PCR positivity was 16% (95% CI: 13–21) in residents and 8% (95% CI: 6–12) in staff. ORs for (re)infection in seropositive (compared to seronegative) residents and staff were 0.22 (95% CI: 0.06–0.72) and 3.15 (95% CI: 1.56–6.63), respectively. (4) Conclusion: during the first wave, RT-PCR test programmes underestimated the number of COVID-19 cases. The reinfection rate in residents was 3%, indicating protection, while it was 21% in staff, potentially due to less cautious health behaviour. Future outbreaks should use both RT-PCR and serological testing for complementary insights into transmission dynamics.

## 1. Introduction

Long-term care facilities, such as nursing homes (NH), have been heavily affected by the COVID-19 pandemic. In Belgium, during the first year of the pandemic, nearly 57% (*n* = 12,447) of all COVID-19 deaths were nursing home residents (NHR). Of these deaths, almost half occurred during the first epidemiological wave, which started on 1 March 2020 and ended on 21 June 2020 [1]. It is now clear that SARS-CoV-2 can rapidly spread once it enters an NH [2,3]. It is therefore important to correctly and promptly identify cases, allowing for timely implementation of preventive measures. Reverse transcription polymerase chain reaction (RT-PCR) is the gold standard for individual patient diagnostics but has certain drawbacks when it comes to epidemiological studies. First, RT-PCR provides an epidemiological snapshot as the result depends on the moment of testing relative to the start of infection [4,5]. Nasopharyngeal sampling performed 10 days after symptom onset decreased the likelihood of a positive test result in the first year of the pandemic [6]. Second, at the beginning of the pandemic, RT-PCR testing capacity was limited and many NHR and nursing home staff (NHS) were not (timely) tested, specifically those who were asymptomatic or only mildly affected [1,7]. Therefore, the true extent of the first wave in NH in Belgium was probably underestimated. By assessing the seroprevalence, a better understanding of the true proportion of infected individuals can be obtained, as SARS-CoV-2 antibodies have been shown to remain detectable up to 12 months after infection in unvaccinated individuals [8]. Seroprevalence can therefore assess the proportion of individuals that developed SARS-CoV-2 antibodies, and hence were infected and might be protected against reinfection, although negative by RT-PCR. Studies in Italy found that during the first and second wave, healthcare workers presented with higher seroprevalence compared to non-healthcare workers. Berseli et al. [8] assessed seroprevalence in healthcare workers in the period of June to September 2020 and found a seroprevalence of 8.8%. Paduano et al. [9] on the other hand found a seroprevalence of 22.9% in healthcare workers in the period of September 2020 to March 2021. Another important factor in tackling transmission involves considering possible reinfection. Knowing if individuals are at risk for reinfection is crucial in infection control. Other respiratory viruses, such as respiratory syncytial virus (RSV) and influenza, can cause reinfections [10]. In SARS-CoV-2, however, reinfection was relatively rare in 2020, despite a world-wide spread [11]. Furthermore, data suggest that natural infection leads to an immune response in most NHR [11] and that seropositive individuals, at least for a certain period, are at a decreased risk for future SARS-CoV-2 infection [12]. Although the pandemic was declared as subsided on 5 May 2023 [13], understanding the pandemic’s impact on NH remains crucial in preparing against potential future outbreaks of SARS-CoV-2 or other infectious diseases. In Belgium, the vaccination campaign in NH started in December 2020 [14]. However, information about the true extent of the first wave and the level of natural protection against severe disease in unvaccinated NHR and NHS is limited. The main goals of our study were (1) to assess the SARS-CoV-2 seroprevalence and compare it with the cumulative RT-PCR positivity during the first epidemic wave in April and May 2020, and (2) to assess the reinfection risk, in unvaccinated NHR and NHS.

## 2. Materials and Methods

### 2.1. Study Design, Setting, and Sample Size

The cross-sectional seroprevalence study was part of a study in which the diagnostic performance of saliva specimens was compared to nasopharyngeal swabs for the detection of SARS-CoV-2 in NHR and NHS in Flanders (Belgium) [15]. That study took place between 30 April 2020 and 29 May 2020. In addition to the collection of saliva and nasopharyngeal swabs, serum was collected in order to gather information about the seroprevalence of SARS-CoV-2. For the main study part (diagnostic performance), a total of 100 participants testing positive by RT-PCR using nasopharyngeal swabs were needed. A total of 1420 participants were included in the main study, of which 1078 provided a serum sample. Serum sampling was only started later in the study; therefore, 342 of the 1420 participants without an available serum sample were excluded. This sample size of 1078 participants was used as a convenience sample for the seroprevalence study described here. In the second part of the study, data about self-reported reinfections were collected retrospectively starting in May 2021, and concerned the period from 30 April until 31 December 2020. These data were collected for all participants of the seroprevalence study (*n* = 1078).

### 2.2. Ethical Approval

The ethical committee of Ghent University Hospital approved the study, BC-07665, on 22 April 2020: reference B.U.N. B6702020000062. The study was conducted according to the approved protocol and the principles outlined in the Declaration of Helsinki. NHS, NHR, and their families were informed by the NH management on the study objectives and sampling procedures. An informed consent form was signed by each participant. If participants were incapable to sign, for example, as with NHR with dementia, the consent form was signed by a confidential counsellor, such as a family member or nurse after approval by the family.

### 2.3. Study Population and Recruitment

#### 2.3.1. Selection of Nursing Homes

The main study was a diagnostic performance study; therefore, participants were not randomized, but a convenience sample was used. Starting from 8 April 2020, the Flemish government organized RT-PCR testing for SARS-CoV-2 in NH [16]. The selection of NH was based on the governmental schedule of planned NH visits for diagnostic testing, which was performed at random. The final selection was based on the willingness and feasibility of the NH management to participate and provide logistical support (e.g., nurses able to assist in the blood sampling) and the feasibility of the UGent study team (e.g., distance). Study visits were performed on the same day as the day of governmental RT-PCR testing.

#### 2.3.2. Selection of Nursing Home Residents and Staff

Unvaccinated NHR and NHS were included if they were adults, and if they or their confidential counsellor were informed on the study goal and agreed to participate. Given the goal of the main study, all interested NHR and NHS were eligible to participate and no randomization of participants was performed. Participants not able to produce a sufficient saliva sample were excluded.

### 2.4. Data Collection

#### 2.4.1. Serum Collection

Venous blood was collected in a serum tube and thereafter labelled with the individual study code. Blood samples were transported the same day to the Laboratory Molecular Microbiology (Ghent University Hospital, Ghent, Belgium), where they were centrifuged at 3000× *g* for 8 min. In cases of clotted blood after leaving at room temperature, the clot was removed, and the tube was centrifuged. The serum samples were stored at −20 °C until analysis. Blood was collected just after nasopharyngeal swab sampling.

#### 2.4.2. SARS-CoV-2 IgG

After thawing sera and vortexing, SARS-CoV-2 IgG antibodies were detected by using a chemiluminescent microparticle immunoassay (CMIA) (Abbott SARS-CoV-2) on the Architect i2000sr Plus system as recommend by the manufacturer. The manufacturer recommended that the cut-off index to define seropositivity was 1.4 (≥1.4 = positive). However, in this study, a cut-off value of 0.9 was used after cut-off optimization.

#### 2.4.3. RT-PCR Testing

Nasopharyngeal swabs for RT-PCR testing taken on the same day as seroprevalence sampling were used to test participants for infection with SARS-CoV-2. Additionally, NH were asked to provide results (both positive and negative) of all RT-PCR tests of participants who provided a serum sample, from March until June 2020 to calculate the cumulative RT-PCR positivity. The results and date of testing were collected after pseudonymization using the individual study code.

#### 2.4.4. Questionnaires

Information about self-reported (re)infection, COVID-19 hospitalization, need for oxygen, and co-morbidities was collected by questionnaires, sent out to the NH in May 2021. Only information on (re)infections in the period between sampling and the date of first vaccination was collected. All questionnaires were pseudonymized. NHS were asked to fill in the questionnaires themselves, while nurses filled in the questionnaires of participating NHR based on their medical file.

### 2.5. Statistical Analysis

Different baseline characteristics were analyzed descriptively. Age was described by median, interquartile range (IQR), and minimum and maximum. Sex and comorbidities were described by relative (%) and absolute (*n*) frequency.

#### 2.5.1. Seroprevalence Analysis

The seroprevalence of SARS-CoV-2 was calculated as the number of positive antibody tests proportional to the total number of participants, and presented with a 95% Wilson score confidence interval (CI). The seroprevalence was calculated per NH, and for all participants included in this study. The data of NH9 were not included in the seroprevalence analysis, because this NH was sampled during an outbreak and all participants were known to have a confirmed SARS-CoV-2 infection.

#### 2.5.2. Cumulative RT-PCR Positivity Analysis

We calculated the cumulative RT-PCR positivity as the sum of all positive RT-PCR tests proportional to the size of the included participants over time. The seroprevalence and RT-PCR positivity were assessed at staggered intervals between 30 April 2020 and 20 May 2020. We assessed all provided results of RT-PCR tests from the start of the pandemic up until the date of seroprevalence sampling of the individual NH. We reported these data per NH and for all NH combined.

#### 2.5.3. SARS-CoV-2 Self-Reported (Re)Infections Analysis

(Re)infection in seropositive and seronegative participants and the treatment of COVID-19 infection (hospitalization and oxygen treatment) were analyzed descriptively. Reinfection was defined as a self-reported (symptomatic or asymptomatic) COVID-19 infection in seropositive participants. The reinfection rate was calculated by dividing the number of reinfected participants by the total number of seropositive participants. Odds ratios and 95% Wilson score CI were calculated to compare the odds of COVID-19 (re)infection in seropositive and seronegative participants. The odds ratio of treatment of COVID-19 infection were not shown due to data scarcity.

Descriptive analyses (median, IQR, minimum, maximum) were performed using Excel (Microsoft Corporation, Redmond, WA, USA) and SPSS (version 26; SPSS inc., Chicago, IL, USA). Missing data were reported for different variables, and complete case analysis was performed because our study is descriptive. One outlier for age (one NHS with an age of 93 years) was excluded. Results did not change after excluding this outlier.

## 3. Results

### 3.1. Participant Flow across the Different Analyses

In Figure 1, the participant flow is displayed. A total of 1078 participants (502 NHR and 576 NHS) from nine NH across Flanders (Belgium) were included. In NH4, participants were tested two weeks apart because of logistic reasons; participants sampled during the second test moment were excluded.

### 3.2. Study Population

The baseline characteristics of all participants (*n* = 1013; 479 NHR and 534 NHS) included in the seroprevalence analysis are shown in Table 1. Four NH were situated in urban and five in rural areas. The study included two public and seven private NH.

### 3.3. Seroprevalence

The seroprevalences across NH and overall are shown in Table 2. Overall, the seroprevalence was 21% (95% CI: 18–23). In NHR, 22% (95% CI: 18–25) had antibodies; in NHS this was 20% (95% CI: 17–24). The seroprevalence per NH varied from 6 to 32%.

### 3.4. Seroprevalence and Cumulative RT-PCR Positivity

Results for cumulative RT-PCR positivity were obtained for five NH (NH1, NH2, NH4, NH7, and NH8) and are presented in Table 2. This subgroup consisted of 314 NHR and 318 NHS. The seroprevalence in these NH was tested between 30 April and 20 May 2020. The overall seroprevalence in these NH at the end of May 2020 was 23% (95% CI: 20–26); in NHR and NHS it was 22% (95% CI: 18–27) and 24% (95% CI: 19–29), respectively. The cumulative RT-PCR positivity on 20 May 2020 was 12% (95% CI: 10–15); in NHR and NHS it was 16% (95% CI: 13–21) and 8% (95% CI: 6–12), respectively. The seroprevalence and cumulative RT-PCR positivity in NHR and NHS were compared descriptively for this sub-cohort overall (Figure 2), and for each NH individually (Appendix A). In three NH, the seroprevalence in NHR was lower compared to the cumulative RT-PCR-positivity (Appendix A).

### 3.5. Self-Reported COVID-19 (Re)Infection in Seropositive and Seronegative Participants

The number of NHR and NHS with self-reported COVID-19 across SARS-CoV-2 IgG-positive and seronegative participants is shown in Table 3. Information about self-reported COVID-19 infection was missing in 22% (18 seropositive and 91 seronegative) NHR and 33% (38 seropositive and 153 seronegative) NHS. Information on treatment (oxygen and hospitalization) in infected participants was missing for two NHR and 14 NHS. The overall reinfection rate in seropositive participants was 10% (95% CI: 7–16), in NHR this was 3% (95% CI: 1–8), and in NHS 21% (95% CI: 13–32). In seronegative participants, we recorded 10% (95% CI: 8–12) primary infections, of which 12% (95% CI: 9–16) were in NHR and 8% (95% CI 5–11) in NHS. Seropositive NHR had lower odds for (re)infection compared to seronegative NHR. In NHS, the odds for (re)infection were higher in seropositive NHS compared to seronegative NHS.

## 4. Discussion

### 4.1. Main Findings

The overall seroprevalence during April and May 2020 in unvaccinated NHR and NHS included in this study was 21% (95% CI: 18–23). The seroprevalence in NHR (22%; 95% CI: 18–25) and NHS (20%; 95% CI: 17–24) was similar. We found a clear discrepancy between prevalence as assessed by means of serology and cumulative RT-PCR. In NH where both were assessed, on 20 May 2020, 24% (95% CI: 19–29) of NHS and 22% (95% CI: 18–27) of NHR presented with SARS-CoV-2 antibodies. However, at that time, cumulative RT-PCR prevalence was only 8% (95% CI: 6–12) for NHS and 16% (95% CI: 13–21) for NHR. The reinfection rate was 3% (95% CI: 1–8) in NHR and 21% (95% CI: 13–32) in NHS, while the primary infection rate in NHR was 12% (95% CI: 9–16) and 8% (95% CI: 5–11) in NHS. Furthermore, we found that seropositive NHR had lower odds for (re)infection compared to seronegative NHR (odds ratio 0.22 (95% CI: 0.06–0.72)). In NHS, the odds for (re)infection were higher in seropositive NHS compared to seronegative NHS (odds ratio 3.15 (95% CI: 1.56–6.63)).

### 4.2. The Seroprevalence in Nursing Homes Was about Three to Four Times Higher Compared to the General Population and Other Healthcare Workers

Compared to the general Belgian population in the same time period, we found a higher seroprevalence in unvaccinated NHR (22%) and NHS (20%). Among blood donors, which can be seen as representatives for the general population, the seroprevalence on 27 May 2020 was estimated to be 5.5% (95% CI: 3.8–7.4) [17,18]. Likewise, when compared to other healthcare workers, the seroprevalence in NHS in our study was higher. Steensels et al. [19] assessed the seroprevalence in 3056 hospital staff in East-Limburg (Belgium) at the end of April 2020. Overall, 6.4% (95% CI: 5.5–7.3) presented with SARS-CoV-2 IgG antibodies. In our study, it was not unexpected to find a higher seroprevalence given the high number of infections and outbreaks in NH in Belgium. During the first wave, 40% of NH reported an outbreak with at least 10 cases [20]. The second epidemiological wave started on 31 August 2020 [21]. During this time, the seroprevalence in NH in Flanders (Belgium) was tested as well in 677 NHR and 508 NHS [22]. The overall seroprevalence was 17% (95% CI: 15–20), and 19% (95% CI: 16–22) in NHR and 15% (95% CI 12–18) in NHS. Although seroprevalence in NHR and NHS possibly decreased over time, we could not find significant differences in seroprevalence between this study and our study, given overlapping CIs.

### 4.3. In the First Epidemic Wave in Flanders, RT-PCR Test Programmes Largely Underestimated the True Infection Prevalence in Nursing Home Residents and Staff

This discrepancy was also seen in Belgian hospital settings. The study of Mortgat et al. [23] assessed the seroprevalence in Belgian hospital staff between April and December 2020. They found that only 56% of seropositive participants had a positive RT-PCR test during and/or before seroprevalence sampling. We believe that an important reason for the discrepancy between the RT-PCR test programme and seroprevalence is the limited testing capacity during the beginning of the pandemic in Belgium [7]. Moreover, many people were asymptomatic and depending on testing strategy, people without or with mild symptoms were often not tested [24]. Furthermore, RT-PCR testing provides only an epidemiological snapshot and cannot detect previous infections that already have been cleared. Last, sensitivity of RT-PCR testing can also be affected by difficulties with the swabbing procedure [4]. Contrary to what was seen in our study in all NH grouped together, in three NH, the seroprevalence in NHR was lower compared to cumulative RT-PCR-positivity. This could partially be explained by a delay in IgG antibody response in the context of a recent outbreak in the NH, where PCR positivity was not yet followed by seropositivity. The study of White et al. [25], who tested antibodies in 669 NHR, found that antibodies were most likely to be detected within 15–30 days of infection. Furthermore, an impaired immune response, especially in older adults, can lead to the absence of detectable antibodies [26].

### 4.4. Seropositive Unvaccinated Nursing Home Residents Had Lower Odds for Reinfection; in Contrast, Seropositive Unvaccinated Staff Presented with Higher Odds for Reinfection

By assessing reinfection rates, we aimed to assess if seropositivity protected against reinfection. Another study in two NH found a similar reinfection rate (1.1%) to the one we found in NHR in our study (3%) [27]. Furthermore, in a prospective study in unvaccinated NHR and NHS in 100 long-term care facilities, a reduced risk of reinfection in participants with SARS-CoV-2 antibodies up to 10 months after primary infection was found [11]. Moreover, seropositivity among healthcare workers not employed in NH was found to be associated with reduced risk of SARS-CoV-2 infection [28,29,30]. Similarly, in the general population, the same trend was seen [31]. In a large Danish population-level observational study, protection against reinfection in RT-PCR-tested individuals was 80.5%; however, in individuals ≥ 65 years it was 47% [31]. Somehow, surprisingly and contrary to findings from other studies, the reinfection rate in NHS in our study was very high (21%). Another study in healthcare workers in the US reported a reinfection rate of about 2% during approximately the same period [32]. In the study of Krutikov et al. [11], antibodies protected against reinfection in both NHR and NHS; however, they found that protection was greater in NHR compared to NHS. The authors hypothesized there to be two possible reasons. First, NHS have a higher availability for RT-PCR testing, because they can also test outside the NH. Second, NHS have higher levels of exposure to infection compared to NHR, both in- and outside the NH [11]. Additionally, we hypothesize that in our study seropositive NHS, after knowing their serology status, possibly adopted a less safe health behaviour compared to seronegative NHS. Furthermore, the risk of a repeated positive test can be increased if primary infection was asymptomatic [33], which can be the case for NHS who were tested systematically and not based on symptoms. Repeated positive tests can occur when there is persistent viral shedding [34]. Even after the resolution of symptoms, RT-PCR can remain positive for several weeks; after 90 days viral shedding is no longer expected [12,35]. We, however, think our results should be interpreted with caution due to the limitations associated with this study, as discussed below.

### 4.5. Strengths and Limitations

In Belgium, information on the seroprevalence during the first wave, during which NH were heavily hit, was lacking. The results of our study indicate an important spread of SARS-CoV-2 within NH during that time. However, this study included certain limitations. First, the study contained a convenience sample of participants. Participants interested in the study or participants who experienced symptoms were possibly more likely to participate, which could lead to selection bias. The selection of the participating NH, however, was based on a list provided by the government and not on any specific characteristics of these NH. Nevertheless, the generalizability might be limited and should be interpreted with caution. Additionally, there can be an underestimation of the seroprevalence due to survivor bias. NHR who died before the time of sampling were not included, which could lead to a selection of a more robust group. Furthermore, there is a potential risk of recruitment bias because we were not able to include NHS who were on sick leave, nor hospitalized NHR. The seroprevalence and RT-PCR positivity in the subgroup of five NH was assessed at staggered intervals between 30 April and 20 May 2020. This limits the comparison to the cumulative RT-PCR positivity. Both an underestimation of RT-PCR positivity and seroprevalence should be considered for those NH sampled earlier in this timeframe compared to those sampled at the end of the timeframe. Additionally, assessments of seroprevalence simultaneously with RT-PCR testing may give an underestimation of antibody levels because antibodies in NHR are most likely to be detected within 15–30 days of infection [25]. In our study 22/72 RT-PCR tests were performed within 14 days before seroprevalence sampling (Appendix A). This was especially the case for NH7, in which 14 of the 15 positive tests were performed within 14 days. For this NH an underestimation of seroprevalence is to be expected. Nevertheless, overall, we saw that seropositivity was higher compared to RT-PCR positivity, which underlines our message that RT-PCR testing underestimated the true extent of the spread of the pandemic in NH during the first epidemiological wave. Furthermore, to assess the risk of (re)infection in seropositive and -negative participants, we used self-reported data without confirmation through RT-PCR testing. This could lead to recall bias. We defined reinfection as a (symptomatic or asymptomatic) infection in seropositive participants; the date of the primary infection was unknown. Furthermore, an important amount of data regarding the questionnaires were missing, including the date of reinfection. Therefore, it is possible that some patients with persistent viral shedding were incorrectly labelled as reinfected.

### 4.6. Practical Implications

One of the crucial challenges in the COVID-19 pandemic was predicting its course. Knowing more about the spread of the virus and the immune response (duration and protection) is crucial. Although the pandemic has subsided, assessing this information can still provide important insight. In this study, we illustrate the limitations that were associated with the RT-PCR test programme in Belgian NH during the first SARS-CoV-2 wave. The RT-PCR test programme failed to detect all cases in NHR and NHS compared to seroprevalence. This underscores the complexity of real-time infection detection during that period. Others have identified different risk factors for transmission in NH: asymptomatic NHS, delayed recognition of symptoms, insufficient testing, and the size of the NH [36,37,38]. However, the most important predictor seemed to be a high prevalence of COVID-19 in the community [36,37,38,39]. At this time, most NHR and NHS have been vaccinated. However, the findings in this study underscore the importance of assessing viral spread during the early stages of the pandemic. The repeated assessment of seroprevalence in (representative samples of) subpopulations adds significantly to epidemiological surveillance and the understanding of the dynamics of (future) outbreaks or pandemics and other infectious diseases. We believe both RT-PCR and serology to play a complementary role in case detection and surveillance.

## 5. Conclusions

During the first epidemic wave, seroprevalence in Flemish NH was 21%. Our study suggests that the RT-PCR test programme during that period largely underestimated the number of COVID-19 cases in NHR and NHS compared to seroprevalence. Furthermore, the reinfection rate in seropositive NHR was 3%, and seropositive NHR seemed to be protected against reinfection. Seropositive NHS, however, presented with a reinfection rate of 21%; this could possibly be due to less prudent health behaviour. We believe that in future outbreaks of infectious disease, the use of both RT-PCR and serological testing can offer complementary insight on transmission dynamics.

## Figures and Tables

**Figure 1 viruses-16-01461-f001:**
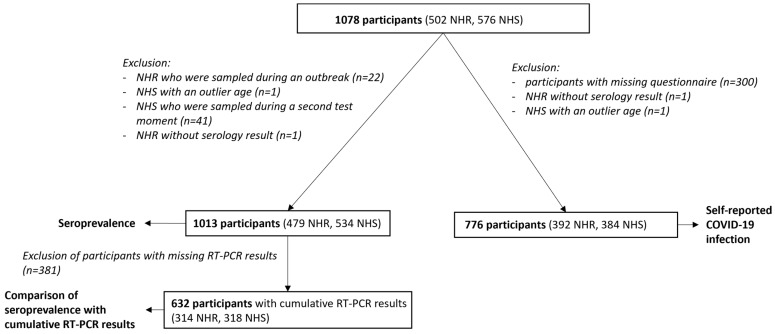
An overview of the total number of participants included in the different analyses in this study. A total of 1078 participants were enrolled in this study. Data used for the seroprevalence analysis were available for 1013 participants, and data for the analysis on self-reported COVID-19 infection were available in 776 participants. Many participants took part in both analyses. NHR, nursing home resident; NHS, nursing home staff; RT-PCR, reverse transcription polymerase chain reaction.

**Figure 2 viruses-16-01461-f002:**
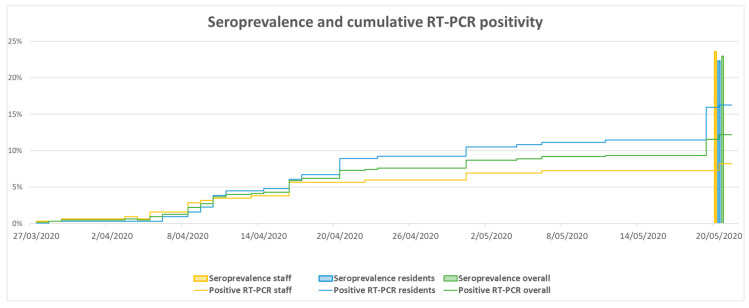
The seroprevalence and cumulative RT-PCR positivity in a selection of nursing home residents (*n* = 314) and staff (*n* = 318).

**Table 1 viruses-16-01461-t001:** Characteristics of participants (*n* = 1013) included in the seroprevalence analysis.

	Nursing Home Residents (*n* = 479)	Nursing Home Staff (*n* = 534)
**Age (years)**		
Median (IQR ^1^ 25–75)	87 (82–91)	40 (30–52)
Minimum–maximum	50–103	16–79 ^2^
Missing data, *n* (%)	2 (<1)	169 (32)
**Sex, *n* (%)**		
Female	325 (68)	454 (85)
Male	154 (32)	80 (15)

^1^ IQR, interquartile range. ^2^ People, often older adults, volunteering in the nursing homes were also included in the study.

**Table 2 viruses-16-01461-t002:** Seroprevalence and cumulative RT-PCR positivity in nursing home residents and staff.

		Seroprevalence			Cumulative RT-PCR ^1^ Positivity		
Nursing Home	Date	Residents, *n*/N (%)	Staff, *n*/N (%)	Overall, *n*/N (%)	Residents, *n*/N (%)	Staff, *n*/N (%)	Overall, *n*/N (%)
Nursing home 1	30/04/2020	25/69 (36%)	20/70 (29%)	45/139 (32%)	10/69 (14%)	7/70 (10%)	17/139 (12%)
Nursing home 2	6/05/2020	21/79 (27%)	20/69 (29%)	41/148 (28%)	5/78 ^2^ (6%)	1/69 (1%)	6/147 ^2^ (4%)
Nursing home 3	7/05/2020	0/80 (0%)	9/74 (12%)	9/154 (6%)	missing	missing	missing
Nursing home 4	11/05/2020	5/26 (19%)	3/16 (19%)	8/42 (19%)	9/26 (35%)	1/16 (6%)	10/42 (24%)
Nursing home 5	13/05/2020	33/84 (39%)	20/99 (20%)	53/183 (29%)	missing	missing	missing
Nursing home 6	15/05/2020	N/A ^3^	4/43 (9%)	4/43 (9%)	missing	missing	missing
Nursing home 7	19/05/2020	9/89 (10%)	11/72 (15%)	20/161 (12%)	15/89 (17%)	0/72 (0%)	15/161 (9%)
Nursing home 8	20/05/2020	10/52 (19%)	21/91 (23%)	31/143 (22%)	12/52 (23%)	17/91 (19%)	29/143 (20%)
Overall	N/A ^3^	103/479 (22%)	108/534 (20%)	211/1013 (21%)	N/A ^3^	N/A ^3^	N/A ^3^
Overall ^4^	N/A ^3^	70/315 (22%)	75/318 (24%)	145/633 (23%)	51/314 (16%)	26/318 (8%)	77/632 (12%)

^1^ RT-PCR, reverse transcription polymerase chain reaction. ^2^ One resident was excluded due to missing results of RT-PCR testing; this resident tested seronegative on antibody testing. ^3^ N/A, not applicable. ^4^ The overall test results for the five nursing homes included in the comparison of seroprevalence and cumulative PCR positivity.

**Table 3 viruses-16-01461-t003:** Self-reported COVID-19 (re)infection in seropositive and -negative nursing home residents and staff.

	** *n* **	**Self-Reported COVID-19 (Re)Infection, *n* (%)**	**Odds Ratio (95% Confidence Interval) ^1^**
**Residents**	392		
IgG+ ^2^	106	3 (3%)	0.22 (0.06–0.72)
IgG−	286	34 (12%)	1
**Staff**	384		
IgG+	70	15 (21%)	3.15 (1.56–6.63)
IgG−	314	25 (8%)	1
	** *n* **	**Oxygen treatment, *n* (%)**	**Hospitalization, *n* (%)**
**Residents ^3^**	37		
IgG+	3	1 (33%)	0
IgG−	34	11 (32%)	2 (6%)
**Staff ^3^**	40		
IgG+	15	0	0
IgG−	25	1 (4%)	1 (4%)

^1^ The odds ratio and 95% Wilson score confidence interval were calculated by comparing the odds of COVID-19 (re)infection in seropositive (IgG+) and seronegative (IgG−) participants, with seronegative participants used as the reference group. Odds ratios were not calculated for treatment (oxygen and hospitalization) due to data scarcity. ^2^ IgG, SARS-CoV-2 Immunoglobulin G. ^3^ Only residents and staff with a self-reported COVID-19 (re)infection were included.

## Data Availability

The datasets used and/or analyzed during the current study are available from the corresponding author on reasonable request.

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
