# Peer review of "SARS-CoV-2 Seropositivity in Nursing Home Staff and Residents during the First SARS-CoV-2 Wave in Flanders, Belgium"

_viruses, 2024, doi:10.3390/v16091461_

Round 1

Reviewer 1 Report

Comments and Suggestions for Authors

The authors (AA) aim to assess the SARS-CoV-2 seroprevalence during the first epidemic wave in April and May 2020 in nursing home residents (NHR) and nursing home staff (NHS), and compare it with the cumulative RT-PCR positivity at that time, and to assess the reinfection risk in NHR and NHS. This is an interesting article useful to increase our knowledge of the issue. Addressing the following issues can make this manuscript eligible for publication.

Introduction

The references are appropriate, but AA could add other references about seroprevalence before and during second wave in other countries such as:

-          Modenese A, Mazzoli T, Berselli N, et al. Frequency of Anti-SARS-CoV-2 Antibodies in Various Occupational Sectors in an Industrialized Area of Northern Italy from May to October 2020. International Journal of Environmental Research and Public Health. 2021; 18(15):7948. https://doi.org/10.3390/ijerph18157948

-          Berselli N, Filippini T, Paduano S, et al. Seroprevalence of anti-SARS-CoV-2 antibodies in the Northern Italy population before the COVID-19 second wave. Int J Occup Med Environ Health. 2022;35(1):63-74. doi:10.13075/ijomeh.1896.01826.

-          Paduano S, Galante P, Berselli N, et al.  Seroprevalence Survey of Anti-SARS-CoV-2 Antibodies in a Population of Emilia-Romagna Region, Northern Italy. Int. J. Environ.  Res. Public Health 2022, 19, 7882. doi: 10.3390/ijerph19137882

Line 73: Specify the period.

Methods

It is not clear whether the serum sample was collected in the same session as the nasopharyngeal swab. Clarify it.

Results

Figure 1: Clarify "self-reported infection". 776 were the respondents to the questionnaire and not those who reported the infection.

Discussion

Was vaccination data included in the questionnaire? How many were vaccinated at the time the questionnaire was administered?

Reviewer 2 Report

Comments and Suggestions for Authors

I thank the authors for their work. Unfortunately, the manuscript contains several significant methodological errors. This means that the conclusions drawn in the manuscript may be incorrect.

First, the study contained a convenience sample of participants. Participants interested in the study or participants who experienced symptoms may have been more likely to participate.

Secondly, the time interval between the collection of nasopharyngeal swabs and blood sampling is not specified. Were these manipulations carried out simultaneously? In this case, the data is not reliable, since antibodies are formed several weeks after infection. Therefore, antibody detection and virus detection cannot be performed simultaneously.

Third, data on reinfection were collected using a survey method. This method is extremely unreliable. Patients may have confused COVID-19 reinfection with another respiratory illness. Therefore, the conclusions about re-infection do not correspond to reality.

Fourth, the manuscript does not contain data on patient vaccination. Were patients in the study group vaccinated? This could also have an impact on reinfection.

Reviewer 3 Report

Comments and Suggestions for Authors

I am grateful for the opportunity to review the manuscript entitled SARS-CoV-2 seropositivity in nursing home staff and residents during the first SARS-CoV-2 wave in Flanders, Belgium. The elderly population is spatially vulnerable, so studies are needed to help us understand the effect of SARS-CoV-2 in this population in order to address their needs and design prevention actions more effectively.

I would like to recommend the following suggestions:

Abstract.

The background should include the objective of the study.

Introduction

Lines 46-50 have the same reference [1]. It is not necessary to repeat the number, it should be put only at the end.

I think this section could be expanded to better contextualise the study. For example, explain the differences between RT-PCR and seroprevalence, why seroprevalence provides a better description of infected individuals (line 59), provide epidemiological data on prevalence in Belgium, explain why reinfection is a factor to be taken into account? And so on.

Section 2.1 is confusing. Of the 1420 participants, a serum sample was obtained from 1078 of them, why is this difference due to the fact that participants without a serum sample were excluded?

In section 2.5.3, how were reinfections identified, with a questionnaire? Are these self-reported reinfections? If so, this could be a reporting bias. Also, how were asymptomatic reinfections identified?

Figure 1: I do not understand the division into 1013 participants seroprevalence and 776 participants self-reported infection, if it is indicated below that many participants had both analyses.

Table 1: in the comorbidity data, too much information is missing (57% for residents and 82% for staff). I believe the information is not representative and should be removed from the table.

Author Response

Please see the attachment for the responses to your comments.

Round 2

Reviewer 2 Report

Comments and Suggestions for Authors

All my questions have been considered. However, the answer to the second question is not satisfactory. Antibodies are formed until several weeks after infection. Antibody detection performed simultaneously with PCR analysis may give a negative result, because antibodies are formed later. Moreover, in some patients, antibodies were detected during the acute period of infection, and in others - after recovery. The antibody titer depends on how much time has passed since infection. Therefore, it is necessary to divide patients depending on the time elapsed since infection and provide antibody analysis data for each group of patients.
